# Natural Products from Plants and Algae for Treatment of Alzheimer’s Disease: A Review

**DOI:** 10.3390/biom12050694

**Published:** 2022-05-12

**Authors:** Jana Klose, Carola Griehl, Steffen Roßner, Stephan Schilling

**Affiliations:** 1Faculty of Applied Biosciences and Process Technology, Anhalt University of Applied Sciences, Bernburger Straße 55, 06366 Koethen, Germany; jana.klose@hs-anhalt.de (J.K.); carola.griehl@hs-anhalt.de (C.G.); 2Paul Flechsig Institute for Brain Research, University of Leipzig, Liebigstraße 19, 04103 Leipzig, Germany; steffen.rossner@medizin.uni-leipzig.de; 3Fraunhofer Institute for Cell Therapy and Immunology, Weinbergweg 22, 06120 Halle, Germany

**Keywords:** Alzheimer’s disease, neurodegeneration, drug development, clinical studies

## Abstract

Neurodegenerative disorders including Parkinson’s disease (PD), Huntington’s disease (HD) and the most frequent, Alzheimer’s disease (AD), represent one of the most urgent medical needs worldwide. Despite a significantly developed understanding of disease development and pathology, treatments that stop AD progression are not yet available. The recent approval of sodium oligomannate (GV-971) for AD treatment in China emphasized the potential value of natural products for the treatment of neurodegenerative disorders. Many current clinical studies include the administration of a natural compound as a single and combination treatment. The most prominent mechanisms of action are anti-inflammatory and anti-oxidative activities, thus preserving cellular survival. Here, we review current natural products that are either approved or are in testing for a treatment of neurodegeneration in AD. In addition to the most important compounds of plant origin, we also put special emphasis on compounds from algae, given their neuroprotective activity and their underlying mechanisms of neuroprotection.

## 1. Introduction

Neurodegenerative diseases are a group of disorders in which neuronal function and survival are seriously affected. Many of these diseases, including Parkinson’s, Huntington’s and Alzheimer’s Disease (AD), are caused by structural changes and the deposition of proteins; therefore, they are also assigned to the group of protein misfolding diseases or amyloidoses [1,2,3]. AD is by far the most common cause of neurodegeneration and dementia. It is estimated that AD currently affects 55 million people worldwide (World-Alzheimer-Report-2021. Available online: https://www.alzint.org/u/World-Alzheimer-Report-2021.pdf, accessed on 4 February 2022). Characteristic symptoms of the disease are progressive memory loss, impaired cognitive function and paranoia. The histopathological hallmarks of AD, extracellular amyloid deposits (“amyloid plaques”), which mainly consist of the peptide Aβ, and intraneuronal neurofibrillary tangles of the hyperphosphorylated protein tau, mainly affect the cerebral cortex and the hippocampus [4,5]. Numerous studies suggest that the disease is initiated by the deposition of Aβ, which starts presumably years or decades before the first symptomatic changes [6]. The slow Aβ deposition triggers a downstream cascade (the amyloid cascade), which involves pathologic tau formation and hyperphosphorylation, widespread neuroinflammation and, finally, neuronal death [7,8]. Although the intense research during the last decades enabled a much better understanding of the crucial events in AD pathogenesis, a curative therapy that halts the progression of the disease is not yet available. Most of the so-called disease-modifying experimental drugs are targeting events of the amyloid cascade such as the generation and aggregation of Aβ and the phosphorylation of tau or the cellular metabolism and energy homeostasis [9]. The drug development in AD is faced with several challenges which has resulted in numerous setbacks in recent years [10]. For instance, the enzymes responsible for Aβ formation also have physiological substrates and functions. This complicates the suppression of amyloid peptide formation without interfering with other proteolytical degradation processes. Prominent examples are the γ-secretase complex and the β-secretase BACE1, which play a role in the formation of Aβ peptides [11,12,13]. Moreover, several reports suggest that Aβ1–40/42 and tau also have physiological functions, which leads one to question whether these represent druggable targets [14,15,16,17]. Also, many of the amyloidogenic proteins are localized in the cell nucleus or cytosol, which makes an effective suppression of the aggregation or the breakdown of the conglomerates, e.g., by antibodies, even more difficult [18]. Third, the efficient passage of the blood-brain barrier is needed and thus the pharmaceuticals are required to meet various physicochemical parameters [19,20]. Hence, methods are currently being examined (e.g., focused ultrasound) to make the blood-brain barrier more permeable [21].

Finally, major factors hampering the development and testing of new drugs are based on the clinical presentation of dementia and the currently available diagnostic biomarkers. AD patients frequently also show the presence of Lewy bodies and thus, significant pathological overlap with patients with dementia with Lewy bodies (DLB). As a result, the clinical testing of new active ingredients does not take place in “pure” Alzheimer’s patient populations. Accordingly, attempts are being made (using imaging methods and genetic analyses, among others) to conduct clinical studies in narrowly defined patient populations at an early stage of the disease [22,23,24]. Previously, numerous approaches were therefore undertaken in patients with a possibly too advanced a disease stage [23,25]. In addition, the available diagnostic biomarkers often do not specifically reflect the neurodegenerative disease or provide enough correlation with the clinical status of the patients. These imponderables could be responsible for the failure of different therapeutic approaches in the clinical phase. As mentioned above, alterations in biomarkers precede the symptoms of the disease [6,26], i.e., the measured value of a biomarker cannot be directly correlated with an effect on cognition. An example of this is the antibody bapineuzumab, which caused a significant change in phospho-tau in CSF in phase 2, but missed clinical endpoints [27].

All of these factors finally led to the numerous failures of disease-modifying drugs in AD clinical trials. The very recent accelerated approval of Aducanumab to treat AD may thus represent a first sign of success. However, the complexity also triggered the intense investigations of other fields, such as drugs from natural sources and nutraceuticals (Table 1). One potential reason is that food supplements may have the status as being generally regarded as safe (GRAS) and thus can be quickly applied in clinical testing, and eventually in combination with experimental drugs. Most of these substances are addressing protective mechanisms to cells by, e.g., anti-oxidative effects. However, there are also compounds in testing which are dedicated to disease-modification by, for example, their influence on immune cells. A prominent example is represented by oligomannate from red algae, which obtained approval for AD therapy in China and is currently being tested in additional clinical trials. Due to the emerging role in clinical testing, this review focuses on the current treatment strategies which are based on natural products. We will review drugs which are currently approved but will put a special emphasis on natural products from algae. 

This review is based on the personal databases and knowledge of the authors. The work was completed by a substantial amount of literature search using the databases PubMed, Google scholar and SciFinder. The database search was performed until end of February 2022. Only articles in which an active compound was isolated were considered. The date of publication was not an exclusion criterion.

## 2. Natural Products from Non-Algal Sources

### 2.1. Esterase Inhibitors

*Galantamine.* The advanced stage of AD is characterized by a widespread loss of cholinergic basal forebrain neurons [28]. The inhibition of the cholinesterases acetylcholinesterase (AChE) and butyrylcholinesterase (BChE) leads to an increased acetylcholine level in the brain [29,30]. 

Galantamine [(4aS,6R,8aS)-5,6,9,10,11,12-Hexahydro-3-methoxy-11-methyl-4aH-[1] benzofuro [3a,3,2-ef] [2]benzazepine-6-ol] (Table 2) was first isolated in 1947 from the common snowdrop *Galanthus nivalis* [31,32]. Later, it was also isolated from *Galanthus woronowii* and the red spider lily, *Lycoris radiata* [32,33,34]. In 1960, it was found that galantamine is an inhibitor of cholinesterase [35]. Due to its activity toward muscle AChE, it was used to treat myopathies, post polio paralytic conditions and neuromuscular blockades after anesthesia [36,37]. In 1977 it was reported that galantamine can reverse the acute anticholinergic syndrome induced by scopolamine [38]. The chemical synthesis of galantamine was upscaled and optimized so that quantities of up to 100 kg could be produced under GMP-conditions in the 1990s [39]. Since 2000, Galantamine has been approved in the USA and Europe for the treatment of the symptoms of AD (for example as Reminyl^®^). It is a reversible, competitive AChE inhibitor and an allosteric modulator of the nicotinic acetylcholine receptors (nAChRs) [40] modulating the α4β2 and α7 nicotinic receptors [41,42,43]. In Phase III studies, it showed side effects like nausea or vomiting with mild severity, mostly during the dose-escalation phase [44]. 

*Huperzine A.* Huperzine A, which is isolated from the Chinese club moss *Huperzia serrata,* is a specific and reversible AChE inhibitor [45]. It binds more tightly and specifically to AChE compared to other inhibitors such as physostigmine, galantamine, donepezil and tacrine [46,47,48]. The dissociation rate from the enzyme is very low [49,50]. The (+)-huperzine A enantiomer and the (−)-huperzine A enantiomer have similar neuroprotective properties, but the (+)-huperzine A enantiomer is 50-fold less potent in inhibiting AChE in an amyloid-β peptide model of toxicity [51]. In another study, the (+)-huperzine A and (−)-huperzine A showed similar results in protecting cells against Aβ toxicity [52]. The neuroprotective effects of huperzine A are created by its potential to protect cells against hydrogen peroxide, β-amyloid toxicity, glutamate, ischemia and staurosporine-induced cytotoxicity and apoptosis [46,47,48,52]. Toxicological studies in different animal species and clinical trials in China have shown that huperzine A has less cholinergic side effects than other AChE inhibitors [47,53,54,55,56]. The most common side effect of huperzine A is nausea [56]. Also, huperzine A improved the memory of aged subjects and patients with AD [54,56]. It is available as a dietary supplement. 

*Physostigmine.* Physostigmine [(3aR,8aS)-1,3a,8-trimethyl-1H,2H,3H,3aH,8H,8aH-pyrrolo [2,3-b]indol-5-ylN-methylcarbamate] is an alkaloid extracted from *Physostigma venenosum* or *Streptomyces pseudogriseolus* [57]. It is the oldest known AChE inhibitor. Physostigmine acts as a pesudosubstrate for BChE and AChE, and the inhibition is the result of a transfer of a carbamate residue onto the active site, which is prone to spontaneous hydrolysis and the recovery of the active enzyme. The inhibition of AChE results in an increased acetylcholine level which leads to stimulation of muscarinic and nicotinic receptors [58,59]. Physostigmine can be used as antidote for the anticholinergic toxicity of antihistamines, atropine, tricyclic antidepressants and phenothiazine [60].

Physostigmine is absorbed in the gastrointestinal tract. The bioavailability ranges between 1–8% [61]. It has a short half-life with a peak plasma concentration after 30 min after oral administration of 2 mg [61,62]. To increase the half-life, the slow release physostigmine salicylate was developed [63,64]. Physostigmine can cause several side effects through indirectly influencing muscarinic receptors which could lead, for example, to nausea, vomiting, diarrhea and abdominal pain and nicotinic receptors which could cause paralysis, muscle twitching and the stimulation of cholinergic receptors in the CNS which could lead to CNS depression [65]. Physostigmine derivatives such as tolserine, eseroline and phenserine were synthesized to improve the short half-life and to prevent side effects. Only phenserine was tested in clinical studies [66].

### 2.2. Plant Natural Products with Antioxidant and Anti-Inflammatory Efficacy

*Ginseng*. Extracts of the rhizome of the plant *Panax ginseng* have been used in Asia for thousands of years to treat different diseases including neurological disorders [67].The extract of the plant has several active compounds, ginsenosides, ginseng polysaccharides, volatile oils, peptides and amino acids [68,69]. There are several ginsenosides identified as useful in the treatment of neurodegenerative disease such as AD, PD and HD. The ginsenoside Rb1, Rg1, Rg2, Rg3, Re and Rh2 and Gintonin showed a beneficial effect on AD symptomatology; Rg1, Re and Rd in PD and Ginseng total saponins and Ginsenosides in HD [70,71,72]. The ginsenosides are classified in two groups: the 20(*S*)-protopanaxadiol (PPD) group and the 20(*S*)-protopanaxtriol (PPT) group. Rb1, Rc, Rb2, Rd and Rg3 belong to the 20(*S*)-protopanaxadiol group, while Rg1, Re, Rg2 and Rh1 belong to the 20(*S*)-protopanaxtriol group [73]. The chemical structure of the ginsenosides is shown in Table 3. Ginsenosides prevent neuroinflammation and oxidative stress. They also have a positive influence on the brain function by apparently diverse mechanisms [74,75,76,77].

For instance, the ginsenoside Rb1 and Rg1 protects spinal cord neurons from oxidative stress induced by H_2_O_2_ and excitotoxicity induced by glutamate and kainic acid with an optimal dose of 20–40 µM [67]. In an AD mouse model, Rg1 showed neuroprotective effects through improved cognition and amyloid pathology, modulation of the amyloid precursor protein process and activation of the hippocampal-dependent protein kinase/hippocampal-respond element-binding protein (PKA/CREB) signalling [78]. The ginsenoside Rb1 has several neuroprotective effects. It promotes neural growth, the expression of growth-promoting kinases and helps prevent their levels from decreasing and has played the role of an antiapoptotic agent after Aβ-induced apoptosis in an AD cell model [79,80]. Furthermore, Rb1 seemed to protect the brain from Aluminium-induced toxicity. It reversed the glycogen synthase kinase 3β and the protein phosphates level and thereby reduced tau phosphorylation [81].

*Ginkgo biloba.* Ginkgo biloba is the oldest living tree species in the world. The standardized Ginkgo biloba extract (GBE) from the dried leaves has neuroprotective effects and is used for the treatment of memory impairment and dementia [83,84]. GBE contains 6% terpenoids, 24% flavonoid glycosides and 5–10% organic acids [85]. The terpenoids include the ginkgolides A, B, C and J (Table 4). Flavonoids and terpenoids are considered to be the pharmacologically active compounds of GBE [86,87]. GBE was shown to reduce the expression of transgenic human amyloid precursor protein expression in mouse brain [88] and to compensate for changes in brain glucose metabolism induced by streptozotocin treatment in rat brain [89].

There are several studies showing a positive effect of GBE on the cognitive function in elderly and AD patients [90,91,92,93]. However, other studies did not show a significant effect in the prevention or treatment of mild cognitive impairment [94,95]. The contradicting outcomes of the studies may be caused by differing compositions of the GBE. The chemical composition depends on the growth conditions and the preparation of the GBE, which highlights the importance to define the composition of drugs derived from natural sources.

### 2.3. Others

*Curcumin* is extracted from the rhizome of the Curcuma species. It is the main compound of the curcuminoids and has shown antioxidant and anti-inflammatory properties [96]. In neurological disorders, curcumin decreased inflammation and ROS. Combined with aerobic yoga, curcumin should improve memory and cognitive function (NCT01811381, Table 1).

The main active compounds in elderberry juice, grape powder and Meganatural-Az grape seed extract are anthocyanins. Anthocyanins have anti-inflammatory and antioxidative properties. In animal models of AD, a neuroprotective activity was observed: anthocyanins extracted from black soybeans reversed d-galactose-, lipopolysaccharide- or Aβ_1–42_-induced oxidative stress and reduced the ROS level [97,98,99,100]. Other anthocyanins inhibited the Aβ- and oxidative stress-induced GSK-3β hyperactivation and hyperphosphorylation of tau protein [101].

*Omega-3 poly unsaturated fatty acids* (PUFAs) are known to reduce inflammation and vascular risk factors. They decrease cell adhesion molecules which could be related to cerebral small vessel disease [102]. Cerebral small vessel disease influences the accumulation of white matter hyperintensities that results in cognitive decline [103]. Also, metabolites showed neuroprotective properties. The ethyl ester icosapent ethyl from Eicosapentaenoic acid (EPA), an omega-3 PUFA, improves the synaptic function and reduces inflammation (Table 5).

*Rapamycin* is a macrolide compound from the bacteria *Streptomyces hygroscopicus*. It inhibits the T and B cell proliferation and was therefore approved by the US Food and Drug Administration (FDA) to suppress the immune system after organ transplantation [104,105,106]. Rapamycin has been shown to reduce Aβ deposition and pathogenic tau phosphorylation to improve synaptic plasticity and to decrease neuroinflammation in mouse models [107,108,109,110,111,112,113].

*Cannabinoids* from THC-free cannabidiol (CBD) oil target the behavioural and psychological symptoms of dementia. The cannabinoid CBD may act via different mechanisms (Table 5). Several studies suggest that it may protect against Aβ-induced and microglia-activated neurotoxicity in vitro, prevent hippocampal and cortical neurodegeneration, reduce tau hyperphosphorylation and regulate microglial cell migration [114,115,116,117,118]. Furthermore, CBD showed anti-inflammatory and antioxidant activities [119]. The anti-inflammatory properties may result from the decrease of inducible nitric oxide synthase (iNOS) and interleukin-1β protein expression [120]. The anti-inflammatory and neuroprotective properties were investigated in a rat model [121].

*Yangxue qingnao* is a traditional Chinese medicine composed of 11 different herbs [122]. It is used to improve the cerebral blood flow and thereby the brain nourishment. In a mouse model of AD, Yangxue qingnao pills improved cognitive deficits and reduced Aβ deposition [122]. They possibly promote the expression of α-secretase and thereby the non-amyloidogenic processing of APP [122].

## 3. Neuroprotective Algal Metabolites

### 3.1. Carbohydrates

Sodium oligomannate is a mixture of oligosaccharides obtained by the depolymerization of alginate from marine brown algae, followed by its oxidation to oligosaccharides [123,124] (Table 6). In November 2019, it was conditionally approved for the treatment of mild to moderate AD in China [125]. The patients treated with sodium oligomannate showed significant improvement in ADAS-cog12 score compared to the placebo group in a phase II study, whereby the treated group did not show significantly more adverse reactions than the placebo group [126]. The mechanism of action is not completely understood. Studies in mice suggest that oligomannate might act via decreasing neuroinflammation by remodeling gut microbiota and balancing the amino acid metabolism, especially phenylalanine and isoleucine [124].

For other carbohydrates from algae, little or no data are available from in vivo studies. In general, the available data support the mainly anti-oxidative and anti-inflammatory properties of these compounds. Many of these carbohydrates are sulphated and thus strongly negatively charged compounds. Carbohydrates stabilize the cell structure and are involved in ion exchange mechanisms [127,128]. Sulphated polysaccharides from *Porphyra haitanesis* exhibited antioxidant activity and inhibited lipid peroxidation in rat liver microsomes [129]. The sulphated carbohydrate porphyran from *Porphyra yezoensis* showed superoxide anion and hydroxyl radical scavenging activity [130]. Sulphated oligosaccharides from the two green algae *Ulva lactuca* and *Enteromorpha prolifera* increased concentrations of glutathione, superoxide dismutase (SOD) and catalase (CAT) [131].

Floridoside (2-*O*-glycerol-α-d-galactopyranoside) extracted from *Laurencia undulata* showed anti-inflammatory activity in LPS-stimulated BV-2 microglia cells (Table 6). Floridoside inhibited the production of NO and ROS and downregulated iNOS and COX-2 on the gene and protein level via inhibiting the phosphorylation of p38 and ERK [132]. Alginate-derived oligosaccharides inhibited LPS/Aβ42-induced NO and PGE2 synthesis, the expression of COX-2 and iNOS and cytokine release. They diminished the TLR4 and NF-κB overexpression in microglial BV-2 cells [133]. Fucoidan, a fucose-containing sulphated polysaccharide, inhibited ROS and TNF-α release [134]. It reduces NO, PGE2, COX-2, iNOS, MCP-1, TNF-α and IL-1β in LPS-stimulated murine BV2 microglial cells. Fucoidan also decreased the phosphorylation of Akt, ERK, p38 MAPK and JNK [135].

Seleno-polymannuronate is a seleno-derivate from polymannuronate which was synthesized from polymannuronate and Na_2_SO_3_ [136]. Polymannuronate is extracted from edible brown algae. Seleno-polymannuronate decreased the production of NO and PGE2 and the expression of COX-2 and iNOS in LPS-treated primary microglia and astrocytes. Sulphated oligosaccharides from the two green algae *Ulva lactuca* and *Enteromorpha prolifera* reduced the levels of IL-6, TNF-α and IFN-γ [131]. κ-Carrageenan oligosaccharides and desulphated derivatives inhibited TNF-α secretion in LPS-activated microglia [137].

### 3.2. Lipids and Proteins

Besides oligosaccharides, lipids have also been described as potential natural products originating from algae that have neuroprotective properties. Hielscher-Michael at al. showed that sulfolipids, membrane components of the thylakoid membrane of microalgae, inhibit the enzyme glutaminyl cyclase (QC). QCs are involved in the formation of pyroglutamate (pGlu)-modified Aβ peptides, whose formation is related to AD pathology [138,139,140]. QC activity is also related to other disorders such as arthritis [141]. QCs catalyse the intramolecular cyclization of *N*-terminal L-glutamine and glutamate residues into pyroglutamic acid. The modified Aβ peptides are no longer degradable by aminopeptidase and accumulate in the brain. Hence, the inhibition of QC is a potential strategy for the treatment of AD [142]. Hielscher-Michael et al. discovered that 22 methanolic extracts with a concentration of 0.2 mg/mL from the algae *Scenedesmus rubescens*, *Scenedesmus producto-capitatus*, *Scenedesmus accuminatus*, *Scenedesmus pectinatus*, *Tetradesmus wisconsinensis* and *Eustigmatos magnus* showed QC inhibitory activity between 15% to 72% [143]. The compounds with QC inhibitory activity were identified as the sulfolipids 1,2-di-*O*-palmitoyl-3-*O*-(6′-deoxy-6′-sulfo-D-glycopyranosyl)-glycerol, 1-*O*-palmitoyl-2-*O*-linolenyl-3-*O*-(6′-deoxy-6′-sulfo-D-glucopyranosyl)-glycerol and 1-*O*-linolyl-2-*O*-palmitoyl-3-*O*-(6′-deoxy-6′-sulfo-D-glucopyranosyl)-glycerol (Table 7) [143].

The glycoprotein of *Undaria pinnatifida* (UPGP) has antioxidant properties through the enhancing of superoxide dismutase (SOD) activity and inhibiting xanthine oxidase (Xox) activity at a concentration of 5 mg/mL and 1 mg/mL [146]. UPGP showed anti-inflammatory properties in LPS-stimulated RAW264.7 macrophages via inhibition of COX-1, COX-2 and NO [146]. UPGP has AChE, BChE and BACE1 inhibitory activities [146]. UPGP inhibited the BACE1 activity in in vitro enzymatic assays [146].

The cyanobacterial peptides tasiamide B and its analog tasiamide F, both isolated from the marine cyanobacterium *Lyngbya* sp., showed BACE-1 (β-site of APP cleaving enzyme) inhibitory activity [144]. Tasiamide B is a more effective inhibitor of BACE-1 [144,145]. It was also extracted from *Symploca* sp., another marine cyanobacterium [145].

### 3.3. Phenols

The bioactive and neuroprotective polyphenols have been typically isolated from brown algae. Typically, they interfere with several signal transduction pathways or function as enzyme inhibitors (Table 8). For instance, eckol, dieckol and 8,8′-bieckol from *Ecklonia cava* showed anti-inflammatory properties in Aβ25–35-stimulated PC12 cells by inhibition of TNF-α, IL-1β and PGE2 synthesis [147]. These phlorotannins further downregulated the proinflammatory enzymes iNOS and COX-2 by interference with the NF-κB pathway [147]. Dieckol suppressed p38, ERK and JNK, while eckol suppressed the activation of p38 and 8,8′-bieckol decreased the phosphorylation of p38 and JNK [147]. In another experiment, dieckol from *Ecklonia cava* suppressed the production of NO and PGE2 and the expression of iNOS and COX-2 in LPS-stimulated murine BV2 microglia. The reduction of IL-1β, TNF-α, NFκB, p38 and ROS was also shown before by others [148]. Antioxidant properties were also observed with diphlorethohydroxycarmalol and 6,6′-bieckol isolated from *Ishige okamurae* [149,150].

Phlorofucofuroeckol B isolated from *Ecklonia stolonifera* lowered the expression of COX-2 and inducible nitric oxide synthase in LPS-stimulated BV-2 cells [151]. It reduced the pro-inflammatory cytokines IL-1β, IL-6 and TNF-α. It prevents the degradation of inhibitor κB-α (IκB-α) and thereby inhibits the activation of NF-κB. Phlorofucofuroeckol B also inhibited the phosphorylation of Akt, ERK and JNK [151]. The phlorotannins phloroglucinol, eckol, dieckol, 7-phloroeckol, phlorofucofuroeckol A and dioxinodehydroeckol from *Eisenia bicyclis* inhibited NO production [152]. Phlorofucofuroeckol A from *Ecklonia stolonifera* attenuated NO, PGE2, iNOS and COX-2 expression [153]. It lowers the level of IL-1β, IL-6 and TNF-α. As Phlorofucofuroeckol B, Phlorofucofuroeckol A prevents the degradation of IκB-α and inhibits thereby the activation of NF-κB. Phlorofucofuroeckol A downregulated JNK, p38 and Akt [153]. 8,8′-bieckol reduced ROS, NO, PGE2, IL-6 and iNOS in LPS-stimulated primary macrophages, RAW264.7 macrophages and LPS-induced septic mice. It lowers the transactivation and NF-κB and nuclear translocation of the NF-κB p65 subunit [154]. 6,6′-bieckol from *Ecklonia stolonifera* attenuated IL-6, NO, PGE2, COX-2 and iNOS in LPS-stimulated BV2 and murine primary microglial cells. It inhibited the transactivation of NF-κB and the nuclear translocation of the NF-κB p65 subunit as well as the phosphorylation of Akt, JNK and p38 MAPK [155]. The phloroglucinol derivatives dibenzo [1,4]dioxine-2,4,7,9-tetratol from *Ecklonia maxima* inhibited AChE [156], while 6,6′-bieckol extracted from the red algae *Grateloupia elliptica* inhibited AChE and BChE [157].

Sargachromenol isolated from *Sargassum micracanthum* decreased NO, PGE2, COX-2 and iNOS and increased IκB-α [158]. Sargaquinoic acid extracted from *Sargassum siliquastrum* showed anti-inflammatory activity trough reducing NO and iNOS, nuclear translocation of NF-κB and JNK1/2 MAPK. It prevents the degradation of IκB-α [159].

Some polyphenols also showed inhibitory activity on esterases. The phlorotannins phloroglucinol, dibenzo [1,4]dioxine-2,4,7,9-tetraol and eckol showed AChE inhibition in in vitro enzyme assays [156]. Dieckol and phlorofucofuroeckol extracted from *Ecklonia cava* inhibited AChE and increased the level of acetylcholine in mice [160].

Sargaquinoic acid and sargachromenol isolated from *Sargassum sagamianum* and *Sargassum serratifolium* and sargahydroquinic acid extracted from *Sargassum serratifolium* showed moderate AChE inhibitory properties and BACE-1 inhibitory activity. Sargaquinoic acid is a potent BChE inhibitor [161,162].

The polyphenols eckol, dieckol, phloroglucinol and dioxinodehydroeckol extracted from *Ecklonia stolonifera* inhibited the self-aggregation of Aβ_25–35_ in vitro [163].

### 3.4. Isoprenoids

Similar to polyphenols, the neuroprotective effect of isoprenoids such as sterols and xanthin derivatives is primarily based on their anti-oxidative radical scavenging and anti-inflammatory properties (Table 9). Numerous studies have been published addressing the antioxidative activity in different, mostly cellular model systems. For instance, the steroid fucosterol extracted from *Pelvetia siliquosa* increased the level of antioxidant enzymes SOD, GPx and CAT and inhibited ROS production [152,168]. It also provided protection from oxidative damage by raising the GSH level and attenuated of the production of iNOS, TNF-α and IL-6, and the phosphorylation of NF-κB, MKK3/6 and MK2 was shown [169,170,171]. Fucosterol from *Panida australis* and *Hizikia fusiformis* reduced IL-1β, IL-6, TNF-α, NO and PGE_2_ in LPS- or Aβ-induced BV2 microglia cells or keratinocytes [172,173]. Fucosterol extracted from the algae *Ecklonia stolonifera*, *Panida australis* and *Sargassum horridum* inhibited AChE and BChE in vitro [172,174,175]. Different types of inhibition were detected depending on the origin. Fucosterol from *Ecklonia stolonifera* showed a selective inhibition of BChE, a non-selective cholinesterase inhibition of AChE and BChE was observed with fucosterol from *Panida australis* and a non-competitive inhibition was detected with the compound from *Sargassum horridum* [172,174,175]. A non-competitive inhibition of the β-secretase BACE1 was observed with fucosterol from *Ecklonia stolonifera* and *Undaria pinnatifida* [176].

The carotenoid fucoxanthin extracted from *Sargassum siliquastrum* prevented H_2_O_2_-induced and reduced ROS-induced DNA damage [177,178]. It also decreased the cytokines IL-6, IL-1β, TNF-α, NO and PGE_2_ and the enzyme activity of COX-2 and iNOS by suppressing the phosphorylation of MAPKs in Aβ_42_-induced BV-2 microglia cells [177]. In the presence of fucoxanthin, enhanced cell survival was observed with LPS-activated BV-2 microglia by activation of the cAMP-dependent signal cascade pathway resulting in the attenuation of the phosphorylation of Akt, NF-κB, ERK, p38 MAPK and AP-1 and reduced levels of TNF-α, IL-6, PGE2, NO and ROS [179]. Fucoxanthin activated the nuclear factor erythroid 2-related factor 2 (Nrf2)/heme oxygenase-1 (HO-1) pathway and increased the secretion of brain-derived neurotrophic factor [179].

Fucoxanthin isolated from *Phaeodactylum tricornutum* inhibited BChE activity in vitro [180]. It possibly interacts with a peripheral anionic site of AChE mediating non-competitive inhibition [183]. Similarly, α-Bisabolol isolated from *Padina gymnospora* inhibited AChE and BChE in vitro [182]. In two other studies, Fucoxanthin suppressed the formation of Aβ1-42 fibrils and oligomers and inhibited Aβ aggregation [184,185]. α-Bisabolol prevents oligomer formation and disaggregates the mature fibrils [186].

Astaxanthin decreased the cytokine levels of IL-6, IL-1β, and TNF-α. It inhibited iNOS, nNOs and COX-2 expression in the hippocampus and prefrontal cortex of male mice [181]. In rats, astaxanthin attenuated NF-κB activity and the expression of IL-1β, TNF-α and the intercellular adhesion molecule 1 [187].

## 4. Conclusions

The recent conditional approval of the monoclonal antibody aducanumab (aduhelm) by the FDA provides a very stimulating signal for all drug development approaches in AD. However, among others, these antibody approaches are still met with doubts about disease modification and safety, as suggested by the decision of the EMA to not provide approval to Aduhelm (Meeting highlights from the Committee for Medicinal Products for Human Use (CHMP) 13–16 December 2021. Available online: https://www.ema.europa.eu/en/news/meeting-highlights-committee-medicinal-products-human-use-chmp-13-16-december-2021, accessed on 2 February 2022). Hence, nutritional approaches and natural products are vital tools for prevention and amelioration of the progression of neurodegeneration. A considerable strength of the natural products is provided by the multifaceted mechanisms of their activity. Prominent examples for that include, for instance, the ginsenosides or the extracts from Ginkgo biloba (GBE), which are currently the subject of late-stage clinical trials (Table 1). The ingredients exert anti-inflammatory and antioxidative properties and have been described to influence the processing of AD-related proteins, providing a multi-pronged molecular approach of intervention. Also, natural compounds are among the first described to address potential novel pathways in neurodegenerative diseases. The most prominent example for that is GV-971 (sodium oligomannate). The currently available data support an influence on the gut microbiome which leads to the amelioration of AD-related symptomatology. The compound is among the first that addresses the “gut-brain-axis”, which has recently become focus of research in neurodegenerative diseases. Besidessodiumoligomannate, the general role of nutrition and nutrient uptake by the digestive tract is further underscored by the recent reports on the LipiDiDiet multinutrient clinical trial in prodromal Alzheimer’s disease [188]. Collectively, the unique properties of these molecules should further encourage the evaluation of combination therapies of, for example, anti-Aβ immunotherapy and treatment with natural products. Because the compounds reviewed here are mostly available without a prescription, a quick introduction into theclinical routine thus appears straightforward.

## Figures and Tables

**Table 1 biomolecules-12-00694-t001:** Natural agents in Clinical trials of Alzheimer’s disease drug development (US National Library of Medicine. Available online: https://clinicaltrials.gov, accessed from September 2021 to November 2021.

Agent	Mechanism of Action	Therapeutic Purpose	Trial Identifier and Status	Phase
Huperzine A	AChE inhibitor, inhibition of Aβ	improve memory	Not yet recruiting NCT02931136	IV
Sodium oligomannate (GV-971)	neuroinflammation modulators, microbiome modulators, amyloid beta-protein inhibitors; reconditioning the dysbiosis of gut microbiota, preventing peripheral immune cells from invading the brain, inhibiting the inflammatory response in the brain targeting protein folding errors in the brain tissue	improve the cognitive function of patients with mild to moderate AD	Recruiting NCT05058040	IV
Sodium oligomannte capsules (GV-971)	neuroinflammation modulators, microbiome modulators, amyloid beta-protein inhibitors; reconditioning the dysbiosis of gut microbiota, preventing peripheral immune cells from invading the brain, inhibiting the inflammatory response in the brain targeting protein folding errors in the brain tissue	improve the cognitive function of patients with mild to moderate AD	Recruiting NCT05181475	IV
Ginkgo biloba	metabolism and bioenergetics; plant extract with antioxidant properties	Improve brain blood flow and mitochondrial function (cognitive enhancer)	Recruiting NCT03090516	III
Sodium oligomannate (GV-971)	reconditioning the dysbiosis of gut microbiota, preventing peripheral immune cells from invading the brain, inhibiting the inflammatory response in the brain targeting protein folding errors in the brain tissue	improve the cognitive function of patients with mild to moderate AD; evaluate safety, tolerability and efficacy of GV-971	Recruiting NCT04520412	III
Curcumin + aerobic yoga	herb with antioxidant and anti-inflammatory properties	decrease inflammation and oxidation related neurotoxicity	active, not recruiting NCT01811381	II
Elderberry Juice	rich in anthocyanins, has anti-inflammatory and antioxidant activity	improve mitochondrial function	completed NCT02414607	II
Grape powder	antioxidant, anti-inflammatory and anticarcinogenic	improves cognitive performance preservation of metabolism in brain regions important to cognitive function	recruiting NCT03361410	II
Icosapent ethyl (IPE)	synaptic plasticity, neuroprotection; purified from of the omega-3 fatty acid EPA	improve synaptic function; reduce inflammation	recruiting NCT02719327	II
Meganatrual-Az Grapeseed Extract	polyphenolic extract with antioxidant properties	anti-oligomerization agent; prevents aggregation of amyloid and tau	recruiting NCT02033941	II
Omega-3 PUFA	fish oil concentrate standardized to long chain in n-3 PUFA content	reduces inflammation and glial activation; enhances amyloid removal; protect small blood vessels	active, not recruiting NCT01953705	II
Rapamycin	anti-inflammatory, antineoplastic; macrolide compound from *Streptomyces hygroscopicus*	selectively blocks the transcriptional activation of cytokines	recruiting NCT04629495	II
Rifaximin	inflammation, infection and immunity; antibiotic	reduce proinflammatory cytokines secreted by harmful gut bacteria	completed NCT03856359	II
Tacrolimus	tau proteins; macrolide from culture broth of a strain of *Streptomyces tsukubaensis*	reduce pathological changes of tau proteins	withdrawn NCT04263519	II
THC-free CBD Oil	anti-oxidant and anti-inflammatory; cannabinoids	behavioural and psychological symptoms of dementia (BPSD) decrease with use of cannabinoids	recruiting NCT04436081	II
VGH-AD1	undisclosed; traditional Chinese herbal medicine	undisclosed (cognitive enhancer)	not yet recruiting NCT04249869	II
Yangxue Qingnao pills	blood circulation; traditional Chinese medicine, composed of Angelicae Sinensis Radix, Chuanxiong Rhizoma, Paeoniae Radix Alba, Rhemannia glutinosa, Uncaria macrophylla Wall, Caulis spatholobi, Spica Prunellae, Catsia tora Linn, Mater Margarita, Corydalis ambigua and Asarum sieboldii	improve cerebral blood flow and brain nourishment	not yet recruiting NCT04780399	II
BDPP (bioactive dietary polyphenol preparation)	metabolism and bioenergetics, amyloid; combination of grape seed polyphenolic extract and resveratrol	prevents amyloid and tau aggregation	recruiting NCT02502253	I
Pomace olive oil	prevent inflammation; lipophilic minor components	consumption of olive oil reduces activation of microglia by TRL (triglyceride-rich lipoproteins)	completed NCT04559828	not applicable
Extra virgin olive oil “Coratina”	anti-amyloid; biophenol	improve cerebral performance	not yet recruiting NCT04229186	not applicable

**Table 2 biomolecules-12-00694-t002:** Chemical structures and characteristics of esterase inhibitors.

Name	Structure	Source	Characteristics	Ref.
galantamine	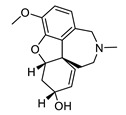	*Galanthus nivalis*	reversible, competitive AChE inhibitor, allosteric modulator of nicotinic acetylcholine receptors, modulates α4β2 and α7 nicotinic receptors	[40,41,42,43]
huperzine A	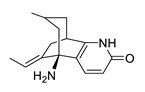	*Huperzia serrata*	specific and reversible AChE inhibitor, protects cells against hydrogen peroxide, β-amyloid toxicity, glutamate, ischemia and staurosporine-induced cytotoxicity and apoptosis	[45,46,47,48,51]
physostigmine	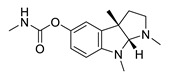	*Physostigma venenosum, Streptomyces pseudogriseolus*	AChE inhibitor	[57]
tolserine	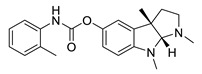	Physostigmine derivative	AChE inhibitor	[66]
eseroline	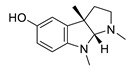	Physostigmine derivative	AChE inhibitor	[66]
phenserine	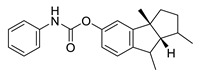	Physostigmine derivative	AChE inhibitor	[66]

**Table 3 biomolecules-12-00694-t003:** Chemical structures of ginsenosides [82].

Structure	Ginsenoside		R1	R2	R3
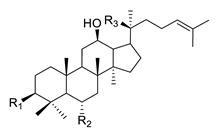	PPD-type	Rb1	-O-Glc-Glc	-H	-O-Glc-Glc
		Rb2	-O-Glc-Glc	-H	-O-Glc-Ara(p)
		Rc	-O-Glc-Glc	-H	-O-Glc-Ara(f)
		Rd	-O-Glc-Glc	-H	-O-Glc
		Rg3	-O-Glc-Glc	-H	-OH
		F2	-O-Glc	-H	-O-Glc
		Rh2	-O-Glc	-H	-OH
		Compound K	-OH	-H	-O-Glc
		PPD	-OH	-H	-OH
	PPT-type	Re	-OH	-O-Glc-Rha	-O-Glc
		Rf	-OH	-O-Glc-Glc	-OH
		Rg1	-OH	-O-Glc	-O-Glc
		Rg2	-OH	-O-Glc-Rha	-OH
		Rh1	-OH	-O-Glc	-OH
		F1	-OH	-OH	-O-Glc
		PPT	-OH	-OH	-OH

**Table 4 biomolecules-12-00694-t004:** Chemical structures of ginkgolides [86,87] from GBE extracts. GBE has been described to reduce APP expression and to improve cognitive function [88,90,91,92,93].

Name	Structure	Name	Structure
ginkgolide A	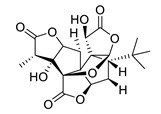	ginkgolide B	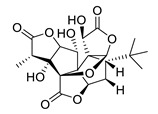
ginkgolide C	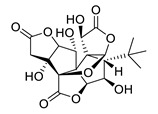	ginkgolide J	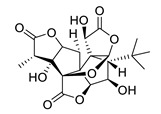

**Table 5 biomolecules-12-00694-t005:** Chemical structures and neuroprotective characteristics of plant natural products from different origin.

Name	Structure	Characteristics	Ref.
curcumin	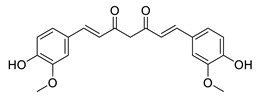	antioxidant, anti-inflammatory, decreases inflammation and ROS	[96]
icosapent ethyl	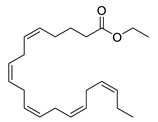	improves synaptic function, reduces inflammation	[103]
rapamycin	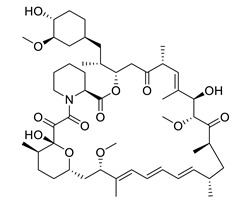	reduces Aβ deposition and pathogenic tau phosphorylation, improves synaptic plasticity, decreases neuroinflammation	[107,108,109,110,111,112,113]
cannabidiol	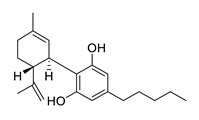	may protects against Aβ-induced and microglia-activated neurotoxicity in vitro, prevents hippocampal and cortical neurodegeneration, reduces tau hyperphosphorylation, regulates microglial cell migration, anti-inflammatory, antioxidant	[114,115,116,117,118,119,120,121]

**Table 6 biomolecules-12-00694-t006:** Chemical structures and neuroprotective characteristics of carbohydrates from algae.

Name	Structure	Source	Characteristics	Ref.
GV971 (Sodium oligomannate)	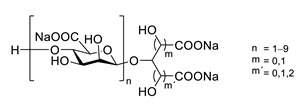	marine brown algae	might act via decreasing neuroinflammation by remodeling gut microbiota and balancing the amino acid metabolism, especially phenylalanine and isoleucine	[124]
porphyran	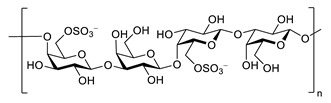	*Porphyra yezoensis*	superoxide anion and hydroxyl radical scavenging activity	[130]
floridoside	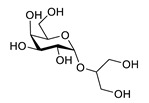	*Laurencia undulata*	anti-inflammatory activity, inhibits the production of NO and ROS, downregulates iNOS and COX-2	[132]
fucoidan	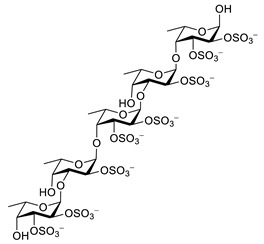	*Ascophyllum nodosum*	inhibits ROS and TNF-α release, reduces NO, PGE2, COX-2, iNOS, MCP-1, TNF-α and IL-1β	[134,135]
κ-carrageenan	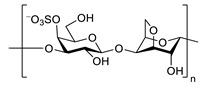		inhibits TNF-α secretion	[137]

**Table 7 biomolecules-12-00694-t007:** Chemical structures, sources and neuroprotective characteristics of lipids and peptides from algae.

Name	Structure	Source	Characteristics	Ref.
1,2-di-*O*-palmitoyl-3-*O*-(6′-deoxy-6′-sulfo-d-glycopyranosyl)-glycerol	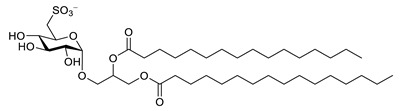	*Scenedesmus rubescens, Scenedesmus producto-capitatus, Scenedesmus accuminatus, Scenedesmus pectinatus, Tetradesmus wisconsinensis*, *Eustigmatos magnus*	QC inhibitor	[143]
1-*O*-palmitoyl-2-*O*-linolenyl-3-*O*-(6′-deoxy-6′-sulfo-d-glucopyranosyl)-glycerol	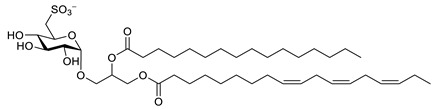		QC inhibitor	[143]
1-*O*-linolyl-2-*O*-palmitoyl-3-*O*-(6′-deoxy-6′-sulfo-d-glucopyranosyl)-glycerol	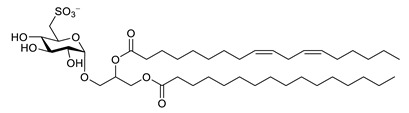		QC inhibitor	[143]
tasiamide B	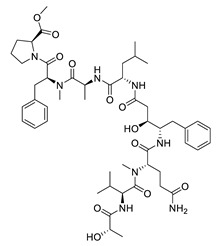	*Lyngbya* sp., *Symploca* sp.	BACE-1 inhibitor	[144,145]
tasiamide F	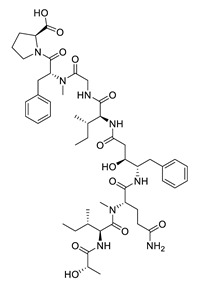	*Lyngbya* sp.	BACE-1 inhibitor	[144]

**Table 8 biomolecules-12-00694-t008:** Chemical structures and characteristics of phenolic compounds from algae.

Name	Structure	Source	Characteristics	Ref.
(−)-cartilagineol	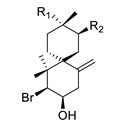	*Laurencia dendroidea*	R_1_ = Cl; R_2_ = Br AChE inhibitor	[164]
(−)-dendroidol			R_1_ = OH; R_2_ = Cl AChE inhibitor	[164]
(−)-elatol	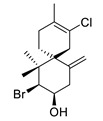	*Laurencia dendroidea*	AChE inhibitor	[164]
2,3,6-tribromo-4,5-dihydroxybenzyl alcohol	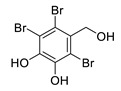	*Symphyocladia latiuscula*	AChE inhibitor, BChE inhibitor	[165]
2,3,6-tribromo-4,5-dihydroxybenzyl methyl ether	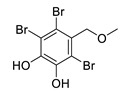	*Symphyocladia latiuscula*	AChE inhibitor, BChE inhibitor, BACE-1 inhibitor	[165]
6,6′-bieckol	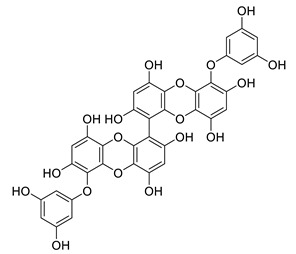	*Ecklonia stolonifera*	decreases of IL-6, NO, PGE2, COX-2 and iNOs	[155]
		*Grateloupia elliptica*	AChE inhibitor, BChE inhibitor, BACE-1 inhibitor	[157]
8,8′-bieckol	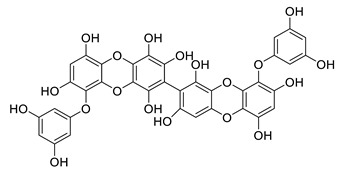	*Ecklonia cava*	inhibits TNF-α, IL-1β and PGE2, downregulates iNOS and COX-2, suppresses p38 and JNK	[147]
			suppresses ROS, NO, PGE2, IL-6 and iNOS, inhibits NF-κB, Akt, JNK and p38 MAPK	[154]
bis-(2,3,6-tribromo-4,5-dihydroxybenzyl) ether	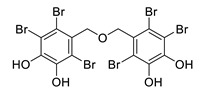	*Symphyocladia latiuscula*	AChE inhibitor, BChE inhibitor, BACE-1 inhibitor	[165]
dibenzol [1,4]dioxine-2,4,7,9-tetraol	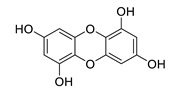	*Ecklonia maxima*	AChE inhibitor	[156]
dieckol	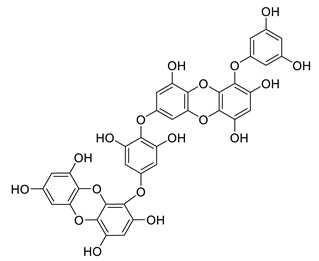	*Ecklonia cava*	inhibits TNF-α, IL-1β, PGE2 and ROS, downregulates iNOS and COX-2, suppresses p38, ERK, JNK and NO, AChE inhibitor	[147,148,160]
		*Eisenia bicyclis*	inhibits NO	[152]
		*Ecklonia stolonifera*	inhibits Aβ_25–35_ self-aggregation	[163]
dioxinodehydroeckol	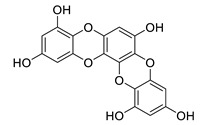	*Eisenia bicyclis*	inhibits NO	[152]
		*Ecklonia stolonifera*	inhibits Aβ_25–35_ self-aggregation	[163]
diphlorethohydroxycarmalol	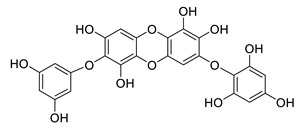	*Ishige okamurae*	antioxidant properties	[149,150]
eckmaxol	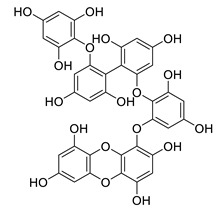	*Ecklonia maxima*	prevents Aβ-induced neuronal apoptosis, decreases ROS	[166]
eckol	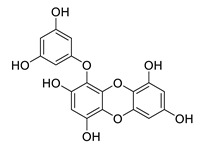	*Eisenia bicyclis*	inhibits NO	[152]
		*Ecklonia stolonifera*	inhibits Aβ_25–35_ self-aggregation	[163]
		*Ecklonia maxima*	AChE inhibitor	[156]
fucofuroeckol-B	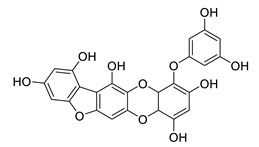	*Eisenia bicyclis*	inhibits β-secretase, attenuates Aβ-induced cytotoxicity	[167]
7-phloroeckol	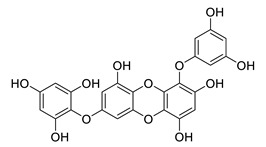	*Eisenia bicyclis*	inhibits NO	[152]
phlorofucofuroeckol A	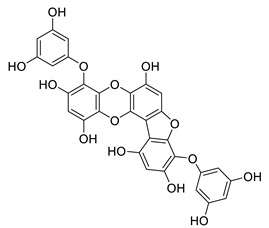	*Eisenia bicyclis*	inhibits NO	[152]
		*Ecklonia stolonifera*	inhibitsNO, PGE2, iNOS, COX-2, IL-1β, IL-6 and TNF-α, increases IκB-α, downregulates NFκB, JNK, p38 and Akt, inhibits Aβ_25–35_ self-aggregation	[153,163]
phlorofucofuroeckol B	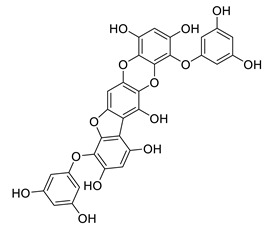	*Ecklonia stolonifera*	downregulates COX-2 and NO, reduces IL-1β, IL-6 and TNF-α, inhibits NF-κB, Akt, ERK and JNK, increases IκB-α	[147,151]
phloroglucinol	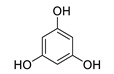	*Eisenia bicyclis*	inhibits NO	[152]
		*Ecklonia stolonifera*	inhibits Aβ_25–35_ self-aggregation	[163]
sargachromenol	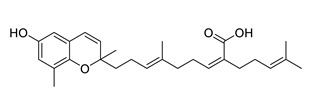	*Sargassum micracanthum*	decreases NO, PGE2, COX-2 and iNOS, increases IκB-α	[158]
		*Sargassum sagamianum*	moderate AChE inhibitor	[161]
		*Sargassum serratifolium*	moderate AChE inhibitor, BACE-1 inhibitor	[162]
sargaquinoic acid	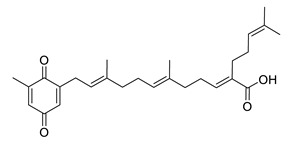	*Sargassum siliquastrum*	reduces NO and iNOS, inhibits NF-κB and JNK1/2 MAPK, increases IκB-α	[159]
		*Sargassum sagamianum*	moderate AChE and BChE inhibitor	[161]
		*Sargassum serratifolium*	moderate AChE inhibitor, BACE-1 inhibitor	[162]
sargahydroquinic acid	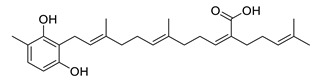	*Sargassum serratifolium*	moderate AChE inhibitor, BACE-1 inhibitor	[162]

**Table 9 biomolecules-12-00694-t009:** Chemical structures and characteristics of isoprenoids from algae.

Name	Structure	Source	Characteristics	Ref.
fucosterol	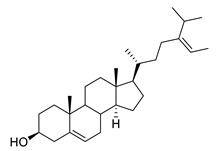	*Pelvetia siliquosa*, *Panida australis*, *Hizikia fusiformis*, *Ecklonia stolonifera*, *Sargassum horridum*, *Undaria pinnatifida*	increases the level of antioxidant enzymes SOD, GPx and CAT, inhibits ROS production, AChE inhibitor, BChE inhibitor, BACE-1 inhibitor	[152,168,169,170,171,172,173,174,175,176]
fucoxanthin	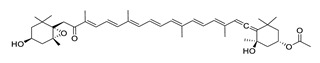	*Sargassum siliquastrum*, *Phaeodactylum tricornutum*	decreases cytokines, prevents H_2_O_2_-induced and reduces ROS-induced DNA damage, inhibits BChE in vitro	[177,178,179,180]
astaxanthin	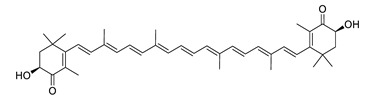		decreases cytokines, inhibits nNOs, iNOS and COX-2 expression	[181]
α-bisabolol	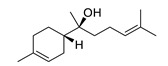	*Padina gymnospora*	inhibits AChE and BChE in vitro	[182]

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
