# Peer review of "Natural Products from Plants and Algae for Treatment of Alzheimer’s Disease: A Review"

_biomolecules, 2022, doi:10.3390/biom12050694_

Round 1

Reviewer 1 Report

The manuscript has an interesting idea, but its development needs a lot of improvement.
Reading is often tiring and unattractive.

There are studies already published on the same topic, which are more dense and bring a better organized compilation of data.

The lack of description of the methodology used is a failure that I consider very serious. What type of revision is the text? What inclusion and exclusion criteria? What types of studies were used (the abstract has the keyword 'clinical studies', but I didn't detect any clinical studies in the text), among many other problems.

Author Response

We feel truly sorry that the initial submission was not convincing to the reviewer and we are taking this criticism seriously. To address the mentioned points, we included a paragraph describing the methodology and databases used for this work at the end of the abstract. Secondly, as this was also criticism mentioned by the other reviewers, we included additional tables to the review, which include chemical structures and keywords on efficacy of the compounds. This should provide a chance for rapid reception of the most important information.

The primary aim of the review was to provide an overview about the mechanism of activity of different natural compounds. Hence, we felt that the information provided in table 1, together with that provided in the text on the mechanisms of action of those compounds, are comprehensively reviewing the information on clinical development stage. 

Reviewer 2 Report

It is really a pity that as a review on natural products, I only see one structure, which is badly drawn. Please draw all the structures of the mentioned Natural Products from Plants and Algae for treatment of Alzheimer’s Disease, otherwise this review will not have any interests at all.

Author Response

We are grateful to the reviewer for this comment and included the structures as requested. This concern was addressed as follows: The structures are incorporated into seven new tables at those sites, were the compounds are reviewed in the text. We also included main articles and the main mode of activity here, in order to enable a rapid reception of this information.

Reviewer 3 Report

The review entitled "Natural Products from Plants and Algae for treatment of Alzheimer’s Disease: A review" describe the potential of natural products from plants and Algae to treat Alzheimer's disease. The review is informative and a good compilation. It is well written and the tables are good explained. However I missed a little more discussion about the common side effects of the approved drugs. maybe one or two phrases no more with references. I also missed the chemical structures and which will be the pharmacophore in case there is one to exert a particular activity. The authors include the gingenosides chemical structures but no the other potential compounds. 

To know the chemical features will bring a plus to this good review. 

The conclusion could be a little stronger, highlighting the major examples and how can move forward on the plant and alga natural products in the authors' opinion.

Other than that the review is good 

Author Response

We thank the reviewer for this evaluation and the suggestions for an improvement. In order to consider the concerns, we added the chemical structures on a tabular basis at sites of appearance in the text. Also, we included potential side effects, where this information was available.

Finally, we revised the conclusion in order to highlight compounds with a pronounced potential for use as common anti-dementia drugs.

Round 2

Reviewer 1 Report

The manuscript has undergone considerable improvement and adjustment since the previous review. 
It may be considered for publication.

Author Response

We thank this reviewer again for his/her time spent to improve this manuscript.

Reviewer 2 Report

Some of the chemical structures are badly drwn, the authors has to redraw them according to ACS document 1996.

Author Response

In order to address the criticism of this reviewer, we revised all structures included in this manuscript again carefully and used ChemDraw software for these drawings. We feel the resolution of the structures is improved considerably and thank the reviewer for insisting on this request.   

Reviewer 3 Report

The manuscript has improved notably

Author Response

We thank the reviewer again for his/her valuable comments and suggestions to improve our manuscript.